# The Molecular Predictive and Prognostic Biomarkers in Metastatic Breast Cancer: The Contribution of Molecular Profiling

**DOI:** 10.3390/cancers14174203

**Published:** 2022-08-30

**Authors:** Benjamin Verret, Michele Bottosso, Sofia Hervais, Barbara Pistilli

**Affiliations:** 1Medical Oncology Department, Gustave Roussy Cancer Campus, 94800 Villejuif, France; 2INSERM Unit U981, Gustave Roussy Cancer Campus, 94805 Villejuif, France; 3Department of Surgery, Oncology and Gastroenterology, University of Padova, 35122 Padova, Italy

**Keywords:** metastatic breast cancer, molecular profilig, molecular biology, targeted therapy, precision medicine, biomarker

## Abstract

**Simple Summary:**

We propose in this article to review the state of knowledge about predictive and prognosis biomarkers in metastatic breast cancer through the prism of molecular profiling studies.

**Abstract:**

The past decade was marked by several important studies deciphering the molecular landscape of metastatic breast cancer. Although the initial goal of these studies was to find driver oncogenic events to explain cancer progression and metastatic spreading, they have also permitted the identification of several molecular alterations associated with treatment response or resistance. Herein, we review validated (*PI3KCA*, *ESR1*, MSI, *NTRK* translocation) and emergent molecular biomarkers (*ERBB2*, *AKT*, *PTEN*, HRR gene, *CD274* amplification *RB1*, *NF1*, mutational process) in metastatic breast cancer, on the bases of the largest molecular profiling studies. These biomarkers will be classed according the level of evidence and, if possible, the ESCAT (ESMO) classification. Finally, we will provide some perspective on development in clinical practice for the main biomarkers.

## 1. Introduction

Since the earliest studies of cancer biology, breast cancer (BC) has been acknowledged as a heterogeneous disease. Early findings by pathologists pointed to morphological features associated with different outcomes that could predict prognosis [1]. During the latter part of the 20th century, the identification of predictive and prognostic biomarkers in BC was reserved to pathologists, which led to the identification of the hormone-receptors (HR) [2] and the human epidermal growth-factor receptor 2 (HER2) [3] and, therefore, to the development of the corresponding immunohistochemistry staining. On the contrary, the beginning of the 21th century was marked by a considerable international effort to further dissect tumors’ heterogeneity and to define the molecular landscape of cancers [4,5,6,7,8]. Previously based on immunochemistry, genomic discoveries led to a new subgroup classification, with the identification of four molecular subtypes (luminal A, luminal B, basal-like and HER2-enriched) in BC, associated with different outcomes and treatment response [9,10,11,12]. Despite improvements in the classification and management of BC, as well as the development of endocrine therapy [13] and anti-HER2 targeted therapy [14], metastatic BC remains in most cases an incurable disease, which will result in over 600,000 deaths worlwide in 2020 [15]. A series of more recent studies have therefore aimed to define the molecular landscape of metastatic BC, in order to identify the driving molecular events involved in tumor progression and metastatic spread. In particular, four major studies have recently reported the genomic characterization of very large cohorts of patients with metastatic BC (Table 1). First, Razavi et al. [16]. reported the molecular landscape of 1918 tumor sample (1000 metastasis, 918 primary tumor from formalin-fixed paraffin embedded (FFPE) tumor biopsy samples) from mBC patients using a hybridization capture-based next-generation sequencing assay, which analyzes all protein-coding exons of between 341 and 468 cancer-associated genes. Among HR+/HER2−mBC, *TP53*, *ESR1*, *ERBB2*, *ARID1A*, *NF1* and *KMT2D* were more frequently mutated in metastatic setting versus early tumors. Interestingly, authors showed that activating *ERBB2* mutations and *NF1* loss-of-function mutations were more common in endocrine-resistant tumors and were mutually exclusive with *ESR1* mutation. Angus et al. [17] reported the whole genome sequencing of 442 snap-frozen metastatic tissue biopsies from mBC, similar to the prior study; *TP53*, *ESR1*, *NF1* mutations were more frequent in a metastatic setting in HR+/HER2− mBC but also *KTM2C*, *PTEN* and *AKT1*. The authors also found that the APOBEC mutational signature was increased in a metastatic setting, mainly related to prior exposure to endocrine therapy, highlighting effects of systemic treatment on the tumor genome. Bertucci et al. [18] performed whole-exome sequencing of 617 tumor samples (543 metastatic sites and 74 breast tumors, frozen biopsies) from mBC patients. Others studied several genes, including *TP53*, *ESR1*, *GATA3*, *KMT2C*, *NCOR1*, *AKT1*, *NF1*, *RIC8A* and *RB1*, finding that they were more frequently mutated in HR+/HER2− mBC. These cancers also showed an increase in mutational signatures S2, S3, S10, S13 (APOBEC) and S17. In metastatic TNBC, the frequency of somatic biallelic loss-of-function mutations in genes related to homologous recombination DNA repair was increased compared to early TNBC. Finally, Aftimos et al. [19] reported the results of targeted gene sequencing from 381 primary tumor and metastatic pairs of breast cancer patients (FFPE and frozen samples). In line with prior studies, *ESR1*, *PTEN*, *CDH1*, *PI3KCA*, and *RB1* mutations were enriched in a metastatic setting.

All of these studies showed an increase of molecular alteration leading to more biological heterogeneity and reflecting effect of anti-tumor treatment.

However, not all molecular alterations have the same level of evidence of clinical actionability. In order to prioritize potential targets according to their proven therapeutic utility, the European Society of Medical Oncology (ESMO) developed the ESMO Scale for the clinical actionability of molecular targets (ESCAT). This classification is based on six levels of evidence, wherein the highest (tier 1) identifies anomalies suitable for routine clinical use based on prospective data and the lowest (tier X) corresponds to alterations for which there is no evidence for their therapeutic utility [20] (Table 2).

Herein, we review the contributions of molecular profiling in identifying the predictive and prognostic biomarkers in metastatic BC and their clinical relevance in terms of response/resistance to treatments according to the ESCAT classification.

## 2. Biomarkers of Response

According to the ESCAT, only five molecular alterations are associated with BC treatment efficacy with the highest level of evidence (ESCAT I): *ERBB2* amplification, germline *BRCA1/2* mutations, *PI3KCA* mutations, microsatellite instability (MSI) and *NTRK* translocations. Considering that *ERBB2* amplifications are usually assessed by immunochemistry or in situ hybridization and *BRCA1/2* mutations by germline testing, they will not be included in this review.

### 2.1. PIK3CA Mutations (ESCAT IA)

Phosphatidylinositol 3-kinases (PI3Ks) are a family of lipid kinases involved in the cell cycle and cell proliferation. They are divided into four classes based on their structures and substrate specificity. The *PIK3CA* gene, encoding the class I catalytic isoform p110α, is involved in the PI3K/AKT/mTOR pathway [22], and up to 40% of HR+ metastatic BC have a *PIK3CA* mutation.

The efficacy of targeting PIK3CA mutations in patients with HR+/HER2− BC was demonstrated in the randomized phase III SOLAR-1 trial. In this trial, 572 patients pretreated with endocrine therapy were randomized to receive fulvestran plus alpelisib, a PI3K-inhibitor, versus placebo. The addition of alpelisib improved median progression-free survival (PFS) from 5.7 to 11 months in the subgroup of patients with *PIK3CA* mutations. In particular, three hotspot mutations of *PIK3CA* (E542K, E545K/A, H1047R/L) predicted the efficacy of alpelisib, whereas no benefit was described among patients with non-mutated BC [23].

The prognostic impact of *PIK3CA* mutation has been evaluated in several studies without clear conclusions. A recent analysis of 649 mBC patients from SAFIR02 trial (NCT02299999) found that, in HR+/HER2− mBC, *PIK3CA* mutations are associated with poor overall survival (OS) and resistance to chemotherapy [24], as well as in HER2+ metastatic BC, wherein *PIK3CA* mutations are associated with poor outcomes and resistance to HER2-targeted therapy [25]. In contrast, in metastatic triple-negative BC (TNBC), *PIK3CA* mutations are associated with a better outcome with a 10-month increase in median OS, in part due to a higher proportion of PI3K mutations in patients with primary HR+/HER2− BC who become triple-negative upon relapse [24].

### 2.2. MSI and NTRK Fusion Cases (ESCAT IC)

Microsatellite instability (MSI) has led to the first FDA-agnostic drug approval based on a tissue biomarker independent of tumor subtype. Indeed, MSI is predictive of the efficacy of pembrolizumab in different cancer types [26,27]. However, in BC, MSI is a very rare event, occurring in less than 1% of cases, and its prognostic impact is still unclear.

Similarly, *NTRK* fusion can be found in several types of cancer [28], and the spectrum of *NTRK*-rearranged cancers is increasing with the growing use of molecular profiling. In BC, *NTRK* alterations are found almost exclusively in secretory carcinoma, an ultra-rare TNBC subtype characterized by *ETV6–NTRK3* gene fusion [29]. Several *NTRK* inhibitors have been developed, and a high response rate has been reported in a case of BC [30].

### 2.3. AKT Mutations and PTEN Deletion (ESCAT IIA)

*AKT*-activating mutations and *PTEN* deletions are found in 7% of metastatic BC [18], leading to the activation of the PI3K/AKT/mTOR pathway. Despite preclinical observations that *PTEN* loss of function is associated with the efficacy of mTOR inhibitors such as everolimus, *PTEN* mutations or low protein expression did not predict everolimus activity in patients with HR+/HER2− metastatic BC in the TAMRAD and Bolero-2 trials [31,32]. Conversely, in HER2+ metastatic BC with *PTEN* loss or *PI3KCA* mutations, everolimus improved PFS [33], highlighting how a similar molecular alteration may have distinct predictive power depending on the BC subtype.

Various AKT inhibitors have been developed and clinically tested. Given the poor efficacy of AKT inhibitors as monotherapy (except in the rare context of germline *PTEN* mutation [34], such as Cowden’s syndrome), several trials have evaluated the combination of AKT inhibitors with other agents. In two small phase II randomized trials with 140 and 124 patients, the addition of capivasertib [35] or ipatasertib [36] to paclitaxel in a first-line setting for metastatic TNBC was associated with a median improvement in PFS of 5 months in patients with a *PIK3CA* or *AKT* mutation or low *PTEN* status. However, these alterations failed to predict the benefit of ipatasertib in the largest prospective phase III clinical trial, IPATunity130 [37], which enrolled 255 mTNBC patients, so they have not yet attained ESCAT I targetability.

In terms of prognosis, *PTEN* alterations have only been associated with worse outcomes in early HER2+ BC [38] and not in other subtypes [39], while their prognostic value in the metastatic setting is still unknown. Similarly, *AKT* alterations, assessed indirectly by phosphorylated AKT (pAKT) immunochemistry, were not predictive of poor outcomes in several retrospective studies [40,41].

### 2.4. Homologous Repair Deficiency/Somatic Alteration of BRCA1/2 (ESCAT IIIA)

Based on two large phase III clinical trials, two PARP inhibitors, talazoparib and olaparib, are currently approved in germline *BRCA1/2* mutated advanced BC [42,43]. However, somatic alterations in *BRCA1/2* can also be found in about 3% of patients with both early and advanced BC, while germline mutations can affect other genes involved in the homologous repair pathway [18,44,45]. PARP inhibitors have therefore been evaluated in patients with germline or somatic alterations in homologous recombination-related (HRR) genes, with promising results.

First, in a small phase II trial enrolling 13 patients with advanced HER2− BC and non-*BRCA* HRR genes mutations, talazoparib induced a significant responses rate of 25%, mainly in patients with germline *PALB2*, *CHEK2* or *FANCA* mutations [46]. The efficacy of talazoparib in metastatic BC with BRCA1/2 somatic mutation is currently being evaluated in an ongoing clinical trial [47]. Olaparib has also been evaluated in patients with advanced HER2− BC and somatic or germline mutations in HRR genes (except germline *BRCA1/2*) in the TBCRC048 trial. In this study, among 54 patients, the response rate in patients with a somatic *BRCA1/2* mutation was about 50% and the median PFS 6.3 months [48]. Finally, Rucaparib was tested in 41 patients with a wild-type BRCA1/2 germline mutation and in patients with advanced BC with homologous recombination deficiency. Five patients (13.5%) showed clinical benefit, including three patients with high loss of heterozygosity (1 complete response and 2 partial responses), one patient with a somatic BRCA1 mutation (stable disease) and one patient with a somatic BRCA2 mutation (partial response) [49].

In term of prognosis, while the outcomes of metastatic BC in patients with germline *BRCA1/2* mutations do not differ from non-carriers or untested subgroups in large real-life databases [50], the prognostic value of the somatic mutation of HRR genes has not been reported to date.

### 2.5. ERBB2 Mutations (ESCAT IIB)

*ERBB2*-activating mutations can be found in about 4% of HR+/HER2− metastatic BC, mainly in the lobular histotype [51]. In vitro studies have shown that *ERBB2* mutations are associated with resistance to endocrine therapy, indeed in a HER2 mutant breast cancer cell model, authors found that HER2-mutated cells were resistant to estrogen deprivation, similarly to *ESR1* mutation, and sensitive to fulvestran plus neratinib [51]. Recent results from the SUMMIT trial, a randomized phase II trial that enrolled 45 patients, showed an overall response rate of 35% and a median PFS of 8.2 months in patients with HR+/HER2− advanced BC with an activating *HER2* mutation and prior CDK4/6 inhibitor treatment treated with trastuzumab–neratinib and fulvestrant [52]. Further studies are ongoing, but anti-HER2 targeted therapies in this setting appear to be an effective strategy.

### 2.6. Immunotherapy

The only validated biomarker for predicting response to immune checkpoint inhibitors in metastatic TNBC is PD-L1 expression, assessed by immunohistochemistry. Indeed, PD-L1 positivity predicted the efficacy of atezolizumab (SP142, cut-off > 1%) [53] and pembrolizumab (CPS, cut-off > 10%) [54] in combination with first-line chemotherapy for advanced TNBC in two phase 3 trials.

Several studies have shown that a high tumor mutational burden (TMB) predicts the efficacy of immunotherapy in several type of cancer [55,56]. BC is generally considered a “medium” mutational burden [57], although this classification is made difficult by its biological heterogeneity and because molecular profiling studies have not found reproducible results. Bertucci et al. showed that TMB was higher in the metastatic setting compared to the early setting in luminal and TNBC [18]. However, Angus et al. did not identify differences in TMB levels between advanced and early BC and among subtypes [17]. Finally, Aftimos et al. reported a higher TMB in the metastatic setting among luminal and HER2-amplified BC but not in TNBC [19]. These differences may reflect the fact that TMB in the metastatic setting is probably more influenced by previous treatments and tumor progression than intrinsic tumor biology.

Nevertheless, in the TAPUR trial, 28 patients with metastatic BC with high TMB were treated with pembrolizumab monotherapy, reaching a response rate of 21% and a median PFS of 10.6 weeks [58]. However, similar results have been reported in unselected patients with metastatic TNBC [59], making TMB a poor predictive biomarker in BC.

In addition, results from the SAFIR-Immuno trial have recently been reported. In this study, 199 patients were randomized to receive durvalumab vs. chemotherapy as a maintenance treatment for advanced BC [60]. Interestingly, the authors found that *CD274’s* (encoding for PD-L1) gain/amplification was associated with better outcomes for the durvalumab group, with a median overall survival not reached vs. 8.8 months. Similarly, recent data from the GeparNuevo trial reported that *CD274* amplification was predictive of durvalumab efficacy in the neo-adjuvant treatment of early TNBC [61].

Although further validation in larger cohorts is needed and it is not yet clear whether patients with *CD274’s* gain/amplification may have a better outcome independently of immunotherapy, these results have identified a promising molecular biomarker and, to date, the only biomarker of response to immunotherapy in both early and advanced settings.

## 3. Biomarkers of Resistance

### 3.1. ESR1 Mutations (ESCAT IA)

*ESR1* encodes the estrogen receptor α, and activating mutations result in constitutive ligand-independent ER activity. ESR1’s mutations have been widely described among HR+/HER2− metastatic BC [62], particularly as a consequence of exposure to prior endocrine treatment with an aromatase inhibitor [17,18,19]. The predictive value of *ESR1* mutations has recently been prospectively evaluated in the phase III PADA-1 trial [63]. In this trial, 1017 patients receiving aromatase inhibitor and palbociclib as first-line treatment for metastatic HR+HER2− BC were screened at regular intervals for *ESR1* mutation in circulating blood. Patients detected with increased circulating *ESR1* mutation without tumor progression were randomized to remain on current therapy or to switch to fulvestrant (known to be more effective in the presence of *ESR1* mutation) in combination with palbociclib. Median progression-free survival in patients who switched before disease progression was significantly increased (11.9 months vs. 5.7 months). In addition to validating the ctDNA monitoring strategy and early treatment intervention, this study provided a prospective validation of *ESR1* mutation with regards to biomarker aromatase inhibitor resistance and fulvestrant sensitivity.

### 3.2. RB1 (ESCAT IV)

Several phase III clinical trials have shown PFS and OS improvements with CDK4/6 inhibitors combined with endocrine therapy in HR+/HER2− metastatic BC [64,65,66]. This class of drugs is the new gold standard in first-line treatment. However, little is known about the resistance mechanisms to CDK4/6 inhibitors. The *RB1* gene controls the G1-S transition through the transcriptional repression of E2F1-proliferation-related target genes, including the cyclin-dependent kinase CDK4/6. Several studies have shown that *RB1* deleterious mutations are associated with poor outcomes [18,67,68], with a median PFS of less than 4 months on endocrine therapy plus CDK4/6 inhibitors. Beyond the negative predictive value of *RB1* deleterious mutations for CDK4/6 inhibitors, these alterations identify a subset of patients with very poor outcomes, worse than those treated without CDK4/6 inhibitors, which also suggests a prognostic value of *RB1* mutations.

### 3.3. NF1 Mutation (ESCAT IV)

*NF1* mutation is found in about 7% of metastatic BC cases, mainly HR+/HER2− [17,18,19]. *NF1* inhibits Ras activation and, when mutated, leads to the activation of the RAS pathway. Bertucci et al. showed that *NF1* mutation was an independent prognostic factor associated with poor outcome [18]. Interestingly, *NF1* mutation and *ESR1* mutations were mutually exclusive and more frequent in the lobular subtype [69]. Preclinical data also suggest that *NF1* is involved in endocrine-therapy resistance. In term of actionability, despite some positive results of MEK inhibition in NF1-related germ cell tumors [70], none of the five patients with NF1 mutations treated with selumetinib (a MEK inhibitor) showed an objective response in the SAFIR02 trial [18]. Thus, the predictive value of *NF1* is still unclear but may be improved in the near future with the development of a RAS pathway’s inhibitor.

### 3.4. APOBEC Mutagenesis (ESCAT IV)

Beyond single-gene alterations, large molecular profiling studies have provided information on mutational signatures defined several year ago by Alexandrov et al. [57]. Among these signatures, the APOBEC signature was found to be enriched in the HR+/HER2− metastatic BC [17,18]. APOBEC mediates genomic mutations from C to T and has been described as an independent prognostic factor associated with worse outcomes [18]. In vitro studies using lentivirus-based system enable the promotion of APOBEC mutagenesis in breast cancer cells have shown that APOBEC mutagenesis induces resistance to endocrine therapy [71], leading to the idea that targeting APOBEC could overcome resistance to endocrine therapy. Targeting mutational processes remains a major challenge; however, interesting preclinical data suggest the efficacy of DDR-related gene inhibitors in APOBEC mutagenesis [72,73,74], although no clinical data are yet available.

## 4. Discussion

In this review, we have summarized the current role of tumor profiling in breast cancer by considering the clinical impact on key predictive and prognostic molecular biomarkers in metastatic BC. Despite significant progress in the understanding of breast carcinogenesis and tumor progression, we must admit that only a few molecular alterations are currently taken into account in the decision-making process of clinical practice (Figure 1). For this reason, the ESMO guidelines do not recommend the use of the multigene next-generation sequencing (NGS) of tumors in breast cancer [75], since PARP inhibitors are only approved for germline *BRCA1/2* mutations and since the Solar-1 trial enrolled patients selected for certain hot-spot *PI3KCA* mutations that can be detected by targeted sequencing. However, this statement is likely to change in the near future. Indeed, as discussed earlier in this review, several promising predictive and prognostic biomarkers are emerging from studies based on the comprehensive genomic profiling of large cohorts of metastatic BC. In line with these results, ESMO also recommend including mBC patients in molecular screening programs in order to assess targeted therapies. The level of evidence for the use of circulating tumor DNA monitoring, including *ESR1* mutation, is improving, with prospective trials that have longitudinally evaluated the predictive and prognostic role of ctDNA in the metastastic setting, such as the PADA-1 trial [63] or the BIOLTALEE trial [76]. In addition, several ongoing prospective phase III clinical trials are testing earlier treatment switching based on increased ctDNA. The Serena-6 trial evaluates whether AZD9833 (an oral SERD) plus CDK4/6 inhibitor is superior to standard therapy in HR+/HER2− advanced breast cancer with a detectable *ESR1* mutation prior to progression (NCT04964934). Beyond the *ESR1* mutation, TRAK-ER trial is evaluating the early detection of molecular relapse with ctDNA monitoring and treatment with palbociclib plus fulvestrant vs. the standard endocrine therapy in patients with HR+/HER2− early BC (NCT04985266). If these trials confirm the results of the PADA-1 trial, they are likely to introduce molecular profiling into daily care. We have to highlight here that ctDNA and “liquid biopsy” can be a useful tool for mBC. Indeed, all of the previous studies cited a focus on tumor biopsies, whereas, for a large proportion of mBC patients, especially those with HR+/HER2− mBC, metastases can only be localized in bones, leading to the technical failure of molecular analysis. Similarly to *ESR1* mutation assessed with ctDNA, testing for *PIK3CA* mutations in ctDNA is concordant with the testing of tumor tissue and the prediction of alpelisib efficacy [77]. In terms of prognosis, molecular profiling’s results reflect the significant heterogeneity of BC. As we have seen, the same molecular alteration can have opposite prognostic value in different BC subtypes, such as *PTEN* deletion and *PI3KCA* mutations. Nevertheless, some molecular alterations, such as *RB1* mutation, highlight a subgroup of metastatic BC patients with a very poor outcome. As the new antibody conjugates, such as trastuzumab-deruxtecan and sacituzumab–govitecan, have revolutionized the treatment paradigm for advanced BC, showing efficacy beyond their original target [78,79,80,81], and it is likely that the treatment strategy for advanced BC will change rapidly in the future with the identification of patients with difficult-to-treat mBC who may benefit more from the introduction of these drugs earlier in the course of the disease.

## 5. Conclusions

Finally, although molecular profiling for mBC is not yet recommended, we believe that, in the near future, it is likely to be implemented in clinics, given the growing number of promising biomarkers that have the potential to provide guidance for personalized treatment strategies.

## Figures and Tables

**Figure 1 cancers-14-04203-f001:**
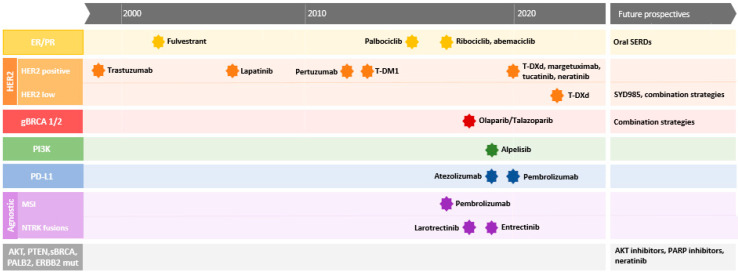
Timeline of main biomarker-based agents approved in metastatic breast cancer. ER, estrogen receptor; PR, progesteron receptor; SERDs, selective estrogen receptor degraders; T-DM1, trastuzumab emtansine; T-DXd, trastuzumab deruxtecan; gBRCA, germaline BRCA; PI3K, phosphoinositide 3-kinase; PD-L1, programmed death-ligand 1; MSI, microsatellite instability; NTRK, Neurotrophic Tyrosine Receptor Kinase; AKT, Alpha-serine/Threonine Kinase; PTEN, Phosphatase and Tensin Homolog; sBRCA, somatic BRCA.

**Table 1 cancers-14-04203-t001:** Largest studies of whole molecular profiling in breast cancer.

Study	Patients	Samples	Subtypes	Alterations Enriched in Metastatic Setting	Sequencing Approach (Depth)
Razavi et al. [16]	1918	1000 metastatic biopsies (purity > 30%) 918 primary biopsie	1364 HR+/HER2− 224 HER2+ 168 TNBC	*TP53* (85%) for TNBC, *NF1* (20%) for ER-/HER2+ BC, *TP53* (25%), *ESR1* (15%), *ERBB2* (5%), *ARID1A* (7%), *NF1* (5%), *KMT2D* (5%) for ER+/HER2− BC	targeted sequencing (MSK-IMPACT, 771×)
Angus et al. [17]	442	442 metastatic biopsies (purity > 30%)	279 ER+/HER2− 77 HER2+ 58 TNBC 28 Unknown	*TP53* (32%), *ESR1* (19%), *NF1* (11%), *KMT2C* (11%), *PTEN* (14%) and *AKT1* (7%), in ER+/HER2− BC	Whole Genome Sequencing (107×)
Bertucci et al. [18]	617	543 metastatic biopsies (purity > 30%) 74 breast tumors	381 ER+/HER2− 30 HER2+ 182 TNBC 24 Unknown	*TP53* (29%), *ESR1* (22%), *GATA3* (18%), *KMT2C* (12%) *NCOR1* (8%), *AKT1* (7%), *NF1* (7%), *RIC8A* (4%) and *RB1* (4%) in ER+/HER2− BC	Whole Exome Sequencing (~20,000 genes, 120×)
Aftimos et al. [19]	381	Primary tumor and metastasis pairs	228 ER+/HER2− 51 HER2+ 71 TNBC 31 Unknown	In the all cohort: *ESR1*, *PTEN*, *CDH1*, *PIK3CA*, and *RB1* mutations; *MDM4* and *MYC* amplifications; and *ARID1A* deletions were enriched	targeted sequencing (>100×)

**Table 2 cancers-14-04203-t002:** ESCAT ESMO Scale for the clinical actionability of molecular targets adapted from Condorelli and Mateo et al. [20,21].

ESCAT Evidence Tier	Level of Evidence			Clinical Implication
	A	B	C	
**I**: Alteration–drug match is associated with improved outcome in clinical trials.	Prospective, randomized clinical trials show that the alteration–drug match in a specific tumor type results in a clinically meaningful improvement of a survival end point.	Prospective, non-randomized clinical trials show that the alteration–drug match in a specific tumor type, results in a clinically meaningful benefit as defined by ESMO MCBS 1.1.	Clinical trials across tumor types or basket clinical trials show clinical benefit associated with the alteration–drug match, with similar benefit observed across tumor types.	Access to the treatment should be considered standard of care.
**II**: Alteration–drug match is associated with antitumor activity but the magnitude of the benefit is unknown.	Retrospective studies show patients with the specific alteration in a specific tumor type experience clinically meaningful benefit with the matched drug compared with alteration-negative patients.	Prospective clinical trial(s) show the alteration–drug match in a specific tumor type results in increased responsiveness when treated with a matched drug; however, no data are currently available on survival end points.	NA	Treatment to be considered ‘preferable’ in the context of evidence collection either as a prospective registry or as a prospective clinical trial
**III**: Alteration–drug match suspected to improve outcome based on clinical trial data in other tumor type(s) or with similar molecular alteration	Clinical benefit demonstrated in patients with the specific alteration (as tiers I and II above) but in a different tumor type; limited/absence of clinical evidence available for the patient-specific cancer type or broadly across cancer types	An alteration that has a similar predicted functional impact as an already studied tier I abnormality in the same gene or pathway but does not have associated supportive clinical data	NA	Clinical trials to be discussed with patients
**IV**: Pre-clinical evidence of actionability	Evidence that the alteration or a functionally similar alteration influences drug sensitivity in preclinical in vitro or in vivo models	Actionability predicted in silico	NA	Treatment should ‘only be considered’ in the context of early clinical trials. Lack of clinical data should be stressed to patients.
**V**: Alteration–drug match is associated with an objective response but without clinically meaningful benefit.	Prospective studies show that targeted therapy is associated with objective responses, but this does not lead to improved outcomes.			Clinical trials assessing drug combination strategies could be considered.
**X**: Lack of evidence for actionability	No evidence that the genomic alteration is therapeutically actionable.			The finding should not be taken into account for clinical decisions.

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
