# Peer review of "The Molecular Predictive and Prognostic Biomarkers in Metastatic Breast Cancer: The Contribution of Molecular Profiling"

_cancers, 2022, doi:10.3390/cancers14174203_

Round 1

Reviewer 1 Report

Dear Editor,

attached please find my comemnts/suggenstions on the Review "The Molecular Predictive and Prognostic Biomarkers in Metastatic Breast  Cancer: the contribution of molecular profiling" by Verret and colleagues.

With Best Regards

Barbara Cardinali

Author Response

Dear reviewer, thank's for your comments, you can find below answer to your comments

1/ We add description of these studies

2/ We have correct it

3/ We have correct it

4/ We have correct it

5/ We have add more information each time it was relevant.

6/  We add a comment on ctDNA in the discussion, anyway because initial recommendation was to write this review for clinician, we have chosen to do not give too much technical details. 

7/ We have correct it

8/ We add this remark

Minor point : We have reviewed the manuscript

Reviewer 2 Report

The manuscript by Verret et al. presents an overview over the recent published literature regarding the prognostic molecular biomarkers in metastatic breast cancer. The manuscript is well structured, it presents each of the main potential molecular biomarkers while also classifying them according to the existing level of evidence. I have the following minor comments:

1. please restructure the abstract in order to highlight which are the prognostic biomarkers reviewed.

2. pay attention to grammar and punctuation. There are multiple sentences that are hard to read due to incorrect wording and multiple typos or joint words. 

Author Response

Dear reviewer, thank's for your comments

1/ We have review the abstract.

2/ The manuscript was reviewed 

Best regards.

Reviewer 3 Report

This review study by Verret et al sheds light on emerging molecular biomarkers in metastatic breast cancer, based on the most comprehensive molecular profiling studies such as ESR1 mutations and endocrine therapy resistance, RB1 deleterious mutations, and primary resistance to CDK4/6 inhibitor or CD274 amplification and sensitivity to immune-checkpoint inhibitor. In this way, these authors suggest that these biomarkers will be compartmentalized according to the level of evidence and if possible, the ESCAT (ESMO) classification. Ultimately, these authors deliver some outlook on evolution in clinical practice for main biomarkers.

I found, that the topic is original and relevant in the field and addresses a specific gap in the field. I believe this review would be very useful for the clinical perspective on metastatic breast cancer.

The methodology is fine and no further control is required.

I found the conclusion to be in line with the evidence and arguments presented.

The references are well updated.

Minor points:

First, it's hard to digest a review article without a figure.

Page no 10, line 235; there is a typo. Authors should fix it.

In Table 1, provide the references. For example, Angus et al17

Table 2 is very difficult to understand text-wise. Authors should fix it.

Author Response

Dear reviewer, thank's for your comments, please find below answers to your comments

First, it's hard to digest a review article without a figure. We have added a figure

Page no 10, line 235; there is a typo. Authors should fix it. We have correct it

In Table 1, provide the references. For example, Angus et al17 We have correct it

Table 2 is very difficult to understand text-wise. Authors should fix it. We have review this table.

Round 2

Reviewer 1 Report

Dear Authors,

attached please find the comments/suggestions to your revised manuscript

Best Regards

Author Response

Dear reviewer,

Thanks for your comments; we have correct references, note that ESR1 is ESCAT IA since results of the prospective trial PADA-1.

We have add some technical details for preclinical studies cited in the manuscript.
Best regards.